# PI3K/AKT/β-Catenin Signaling Regulates Vestigial-Like 1 Which Predicts Poor Prognosis and Enhances Malignant Phenotype in Gastric Cancer

**DOI:** 10.3390/cancers11121923

**Published:** 2019-12-03

**Authors:** Bo-Kyung Kim, Jae-Ho Cheong, Joo-Young Im, Hyun Seung Ban, Seon-Kyu Kim, Mi-Jung Kang, Jungwoon Lee, Seon-Young Kim, Kyung-Chan Park, Soonmyung Paik, Misun Won

**Affiliations:** 1Personalized Genomic Medicine Research Center, KRIBB, Daejeon 34141, Korea; imjy@kribb.re.kr (J.-Y.I.); seonkyu@kribb.re.kr (S.-K.K.); kmj87@kribb.re.kr (M.-J.K.); kimsy@kribb.re.kr (S.-Y.K.); kpark@kribb.re.kr (K.-C.P.); 2Department of Surgery, Yonsei University College of Medicine, Seoul 03722, Korea; JHCHEONG@yuhs.ac.kr; 3Serverance Biomedical Science Institute, Yonsei University College of Medicine, Seoul 03722, Korea; SOONMYUNGPAIK@yuhs.ac.kr; 4Biotherapeutics Translational Research Center, KRIBB, Daejeon 34141, Korea; banhs@kribb.re.kr; 5Immunotherapy Convergence Research Center, KRIBB, Daejeon 34141, Korea; jwlee821@kribb.re.kr; 6KRIBB School of Bioscience, Korea University of Science and Technology, Daejeon 34113, Korea

**Keywords:** gastric cancer, MMP9, PI3K, VGLL1

## Abstract

Although gastric cancer is a common cause of cancer mortality worldwide, its biological heterogeneity limits the available therapeutic options. Therefore, identifying novel therapeutic targets for developing effective targeted therapy of gastric cancer is a pressing need. Here, we investigate molecular function and regulatory mechanisms of *Vestigial-like 1* (*VGLL1*) in gastric cancer. Microarray analysis of 556 gastric cancer tissues revealed that VGLL1 was a prognostic biomarker that correlated with PI3KCA and PI3KCB. *VGLL1* regulates the proliferation of gastric cancer cells, as shown in live cell imaging, sphere formation, and in vivo xenograft model. Tail vein injection of NUGC3 cells expressing sh*VGLL1* resulted in less lung metastasis occurring when compared to the control. In contrast, larger metastatic lesions in lung and liver were detected in the *VGLL1*-overexpressing NUGC3 cell xenograft excision mouse model. Importantly, *VGLL1* expression is transcriptionally regulated by the PI3K-AKT-β-catenin pathway. Subsequently, MMP9, a key molecule in gastric cancer, was explored as one of target genes that were transcribed by VGLL1-TEAD4 complex, a component of the transcription factor. Taken together, PI3K/AKT/β-catenin signaling regulates the transcription of *VGLL1*, which promotes the proliferation and metastasis in gastric cancer. This finding suggests VGLL1 as a novel prognostic biomarker and a potential therapeutic target.

## 1. Introduction

Gastric cancer is biologically heterogeneous and it presents various genetic alterations. Most gastric cancer patients are diagnosed at an advanced stage, and conventional chemotherapy has shown limited efficacy. A recent report predicted the prognosis and response to adjuvant chemotherapy in gastric cancer patients by classifying individuals based on cancer-related genes [1]. Although significant efforts have been made for targeted molecular therapies, only a few targeted therapies that extend survival are currently available for patients with gastric cancer. The majority of these patients exhibit resistance while the monoclonal anti-HER2 antibody Trastuzumab is effective in some *HER*-positive gastric cancer patients [2,3,4]. Ramucirumab, which is an antibody that targets VEGFR-2, has proven to be useful alone or in combination with paclitaxel as a second-line treatment for advanced gastric cancer [5,6]. To date, the gastric cancer-related potential therapeutic targets include EGFR, VEGF, MET, FGFR, PI3K/mTOR, and HDAC [5,7,8]. Unfortunately, clinical trials for inhibitors of these identified targets have failed to demonstrate significant clinical efficacy [5,6]. Therefore, there is a critical need to identify novel therapeutic targets based on the molecular mechanisms of gastric cancer.

The Cancer Genome Atlas classifies gastric cancer into four types based on genetic changes [9,10]. Somatic mutations, such as amplification, deletions, and structural abnormalities in chromosomal region, TP53 mutations, and amplification of the RTK/RAS/MAPK pathway characterize the chromosomal instability (CIN) subtype. Genomically stable (GS) subtype, which is associated with diffuse gastric cancer, has frequently mutations in RHOA and CDH1 gene or fusions involving Rho-family GTPase-activating proteins (GAP). The microsatellite instability (MSI) subtype is characterized by *PI3KCA, ERBB3*, and *MLH1* silencing, MHC class 1 gene alterations, and tumor-specific neoantigens. The Epstein-Barr virus (EBV) subtype has a high rate of *PIK3CA* mutations and high *PD-L1*/*L2* expression. *PIK3CA* mutation frequently occurs in 80% of EBV subtype tumors and 3% of CIN subtype tumors [11]. The mutation or amplification of *PIK3CA* and *PIK3CB* promotes cell growth and drug resistance [12,13]. 

The *Vestigial-like* (*VGLL*) family that comprises *VGLL1*-*4* is named after the Drosophila transcriptional coactivator Vestigial (Vg) [14]. They contain the TOUDU domain to mediate interactions with TEA domain transcription factors (TEADs), which are essential in development [14,15]. *TEAD4*, which is a member of the Hippo signaling pathway, is overexpressed in various cancers and it requires coactivators, such as Yes-associated protein (*YAP*) or transcription coactivator with a PDZ-binding motif (TAZ) to induce the expression of *c-myc*, *Axl*, *survivin*, *CTGF*, *cyr61*, and *VEGF-A* [15,16,17]. VGLL4 functions as a tumor suppressor in cancer by competing with YAP for TEAD binding [18,19,20]. Interestingly, the structural similarities between VGLL1 and YAP or TAZ suggest the formation of the VGLL1–TEAD complex [21,22]. *VGLL1* expression is reported to be associated with reduced overall survival (OS) in triple-negative basal-like breast carcinoma [23]. However, the molecular function of *VGLL1* in cancer remains unclear.

Here, we investigated the clinical relevance and molecular function of *VGLL1* in gastric cancer through in vitro experiments and in vivo mouse models. We further explored the underlying regulatory mechanisms for evaluating VGLL1 as a potential therapeutic target in gastric cancer

## 2. Results

### 2.1. VGLL1 Is a Novel Prognostic Biomarker Correlated with PIK3CA in Gastric Cancer

We assessed the clinical relevance of VGLL1 in gastric cancer by immunohistochemistry (IHC) of gastric cancer specimens. In adenocarcinoma tissues, VGLL1 expression was 55% higher when compared to healthy tissues (Figure 1a). VGLL1 expression was high in NUGC3, NCI-N87, SNU16, SNU216, and MKN28 cells (Figure 1b), but it was barely detectable in SNU5, SNU484, SNU668, and Hs746T cells.

Next, we performed microarray analyses of specimens from 556 gastric cancer patients to understand the clinical relevance of VGLL1 [1]. The overall survival (OS) and recurrence-free survival (RFS) rates were lower in the VGLL1-high subgroup than in the VGLL1-low subgroup (Figure 1c,d). The high expression of VGLL1 was significantly associated with Lauren classification, primary tumor (pT), malignancy (TNM), and lymphatic invasion status (Appendix A). These results indicated positive correlations between VGLL1 expression and the clinicopathological parameters in gastric cancer patients.

We explored 172 genes in the subgroup with high VGLL1 expression to gain insights into the role of VGLL1 in gastric cancer. Gene Ontology analysis suggested the significance of phosphatidylinositol 3 kinase signaling, phosphatidylinositol-3-phosphate biosynthetic processes, phosphatidylinositol phosphate kinase activity, and 1-phosphatidylinositol-4-phosphate-3-kinase activity (Figure 1e,f). VGLL1 expression was positively correlated with PIK3CA and PIK3CB, which are associated with poor OS in gastric cancer patients (Figure 1g,h). 

### 2.2. VGLL1 Regulates the Proliferation of Gastric Cancer Cells

We examined the effect of VGLL1 expression on cell proliferation to understand the VGLL1 function. VGLL1 knockdown inhibited the growth of NUGC3, AGS, and NCI-N87 cells (Figure 2a; Appendix A), whereas VGLL1 overexpression enhanced the growth of NUGC3, AGS, HEK293T, SNU484, SNU638, and SNU668 cells (Figure 2b; Appendix A). In a spheroid culture assay, NUGC3 cells stably overexpressing VGLL1 formed larger spheroids than those of the control cells (Figure 2c). In vivo tumor formation of NUGC3 cells expressing VGLL1-specific shRNAs was significantly reduced when compared to the control (Figure 2d). However, the VGLL1-overexpressing HEK293T cells formed larger tumors than the control cells (Figure 2e). Although HEK293T are not gastric cancer cells, these results are suggestive, but they do not prove that VGLL1 is important for the increased proliferation of xenografts.

### 2.3. VGLL1 Regulates Metastasis of Gastric Cancer Cells in in Vitro and in Vivo Mouse Models

We then investigated the role of VGLL1 in cancer metastasis. In a migration assay, NUGC3 cells that were treated with shVGLL1 migrated more slowly than the control cells, whereas VGLL1-overexpressing NUGC3 cells migrated faster than the control cells (Figure 3a). Moreover, the invasiveness of VGLL1-overexpressing NUGC3 cells significantly exceeded that of the control cells (Figure 3b). In an in vivo mouse model, tail vein injection of NUGC3 cells expressing shVGLL1 resulted in fewer lung metastasis, whereas bigger nodules and a larger metastatic area were found in the lungs of mice expressing the shControl (Figure 3c). Furthermore, metastatic lesions were detected by H&E staining and Ki67 IHC in the lungs and liver in a xenograft model with VGLL1-overexpressing NUGC3 cells (Figure 3d). These results suggest that VGLL1 plays a crucial role in gastric cancer metastasis.

### 2.4. PI3K/AKT/β-Catenin Signaling Participates in the Regulation of VGLL1 Expression

Microarray analyses revealed correlations between VGLL1 expression and PIK3CA and PIK3CB levels (Figure 1g). We also found that reducing the expression of PIK3CA or PIK3CB suppressed VGLL1 mRNA levels (Figure 4a). LY294002, a class I PI3K inhibitor, downregulated VGLL1 mRNA and protein levels in a dose-dependent manner in gastric cancer cell lines (Figure 4b–d). Interestingly, LY294002 inhibited the transcription of VGLL1, but not that of VGLL4 or YAP1, indicating the VGLL1-specific regulation by PI3K/AKT signaling (Figure 4d). In addition, LY294002 blocked AKT phosphorylation, thereby suppressing β-catenin (Figure 4e). Importantly, transient expression of constitutively active AKT containing S473D/T308D double mutation rescued VGLL1 expression in the presence of LY294002 (Figure 4f), which suggested that PI3K/AKT signaling regulates VGLL1 transcription in gastric cancer cells.

We performed an analysis of the VGLL1 promote to understand the mechanism underlying the transcriptional activation of VGLL1. The VGLL1 promoter contains the consensus sequences of binding sites for TCF4, LEF1, p300, and ELK1 (Figure 4g). β-catenin forms a complex with LEF/TCF for the transcriptional activation of target genes. The ChIP assay showed that β-catenin bound to the region containing LEF1 and TCF4 binding sites of the VGLL1 promoter. As expected, LY294002 treatment reduced the binding of β-catenin to the VGLL1 promoter, implicating PI3K/AKT/ β-catenin signaling for the transcription of VGLL1 (Figure 4g). VGLL1 expression was regulated by both the knockdown and overexpression of β-catenin (Figure 4h). Promoter activity was inhibited by knockdown of TCF4, LEF1, or p300 (Figure 4i). In contrast, VGLL1 promoter activity and expression were significantly increased with the co-expression of β-catenin/p300 or p300/LEF1 (Figure 4j,k). These results suggested that the β-catenin/p300/TCF4/LEF1 complex regulates VGLL1 expression at the transcription level, which acts downstream of PI3K/AKT.

### 2.5. VGLL1, a TEAD4 Cofactor, Regulates MMP9 Transcription

We conducted a microarray analysis of VGLL1 siRNA-treated NUGC3 cells to understand the downstream regulation of VGLL1 (Appendix A). It seems that VGLL1 regulates various genes that are involved in cell surface receptor-linked signal transduction, immune response, cell adhesion, wound healing, and cell migration (Appendix A).

We selected MMP9 as a potential VGLL1-targeted gene, because high MMP expression is correlated with metastasis and poor prognosis in gastric cancer [24,25]. We found that MMP9 knockdown inhibited the proliferation and metastasis of NUGC3 cells (Figure 5a,b). VGLL1 knockdown reduced the MMP9 mRNA levels in NUGC3, AGS, and NCI-N87 cells (Figure 5c). MMP9 protein level was reduced by VGLL1 knockdown and increased by VGLL1 overexpression (Figure 5d). The MMP9 protein level was increased in metastasis-bearing liver and lung tissue samples in the VGLL1-overexpressing xenograft mouse model (Figure 5e).

We found two TEA sites at positions −1030 and −571, for binding of TEAD4 to the MMP9 promoter (−1295 to +1), and thus generated luciferase reporter systems with MMP9 promoters containing the wild-type TEA, deleted TEA (dTEA), and mutated TEA sites (mTEA at −571) (Figure 5f). The degree of MMP9 promoter activation was similar between the −1295 and −657 constructs, which indicated that the −1030 TEA site is not crucial in the binding of VGLL1 to TEAD4. Deletion or mutation of the TEA site at –571 resulted in significantly reduced MMP9 promoter activity (Figure 5g). TEAD4 knockdown decreased the VGLL1-induced MMP9 promoter activity (Figure 5h), which suggested that MMP9 expression coordinates TEAD4 and VGLL1 activity.

We then assessed whether the interaction between VGLL1 and TEAD occurred at the TEA sites in the MMP9 promoter. Interaction between VGLL1 and TEAD4 was observed by immunoprecipitation assays (Figure 5i). In chromatin immunoprecipitation (ChIP) assay, VGLL1 and TEAD4 both bound to the TEA region (a) but not to the control (b) (Figure 5j). However, in siTEAD4-treated cells, VGLL1 did not bind to the (a) region, indicating that TEAD4 is required for the binding of VGLL1 to the MMP9 promoter. These data indicate that VGLL1 functions as a cofactor of TEAD4, which binds to the TEA site at −571 in the MMP9 promoter.

Next, we asked whether YAP affects MMP9 transcription. The transient overexpression of VGLL1 increased MMP9, but not CTGF mRNA levels (Figure 5k). Likewise, the transient overexpression of YAP increased CTGF but not MMP9 mRNA levels, which suggests that YAP-independent MMP9 expression occurs in NUGC3 cells.

## 3. Discussion

The Hippo signaling pathway suppresses the proliferation and metastasis of cancer cells by negatively controlling YAP and TAZ, which are cofactors of TEAD transcription factors. The structural similarities between VGLL1 and YAP or TAZ suggest the formation of VGLL1–TEAD complex for cancer malignancy [21]. However, the molecular mechanism of VGLL1 in cancer is still poorly characterized. Here, we present the regulation mechanism of *VGLL1* to induce MMP9 expression that promotes gastric cancer malignancy (Figure 6).

Genetic alterations in the *EGFR*, *MET*, *ERBB2*, and the *PI3K*/*AKT* pathway are often observed in gastric cancer. *PIK3CA* is the most mutated PI3K isoform, with an 18% mutation and 5% amplification frequency in gastric cancer [26]. Microarray analysis of gastric cancer patient tissues revealed that *VGLL1* was a prognostic biomarker and its expression was highly correlated with that of *PIK3CA* or *PIK3CB*. Moreover, a high expression of *VGLL1* and *PIK3CA* predicted worse OS in gastric cancer patients.

We observed that VGLL1 promoted the proliferation and tumorigenesis of gastric cancer cells in in vitro cell culture and in vivo xenograft mouse models. Importantly, sh*VGLL1*-expressing NUGC3 cells suppressed lung metastasis in a mouse model, implying a crucial role of VGLL1 in cancer metastasis.

We revealed that PI3K/AKT signaling was involved in the regulation of *VGLL1* transcription. Promoter analysis demonstrated that the β-catenin/p300/TCF4/LEF1 complex induces the transcription of *VGLL1*. MMP9 was explored as a potential candidate target genes of VGLL1, because it has been reported as a molecular marker of metastasis in gastric cancer [24,25,27]. VGLL1 interacted with TEAD4 at the −571 TEA site in the *MMP9* promoter. The MMP9 levels were elevated in metastasized cancer cells in the lungs and liver in a *VGLL1*-overexpressing NUGC3 xenograft model. The MMPs, a family of zinc containing enzymes, degrade components of the extracellular matrix and regulate extracellular matrix turnover, as well as cancer processes, such as proliferation, angiogenesis, and tumor metastasis [27,28]. Interestingly, MMP9 mainly promotes tumor invasion and metastasis, while TIMP-1 inhibits the functions of MMP9 in gastric cancer, which suggests that the imbalance between MMP9 and TIMP-1 expression may occur in tumor progression [25]. We will further investigate the relationship between VGLL1 and TIMP.

Reportedly, the amino acid sequences SVIFT/HQIVHV of β2 at interface I and the VxxHF/LxxLF motif at interface II of VGLL1/YAP interact with TEAD4 to drive the expression of target genes [21,29]. Interestingly, we observed that VGLL1 regulated only *MMP9* mRNA expression, whereas YAP only regulated *CTGF* mRNA expression in NUGC3 cells. Moreover, cell proliferation promoted by *VGLL1* overexpression was not affected by *YAP* overexpression or knockdown (Appendix A), which indicated that VGLL1 and YAP1 independently induced transcription of their target genes. Therefore, it is likely that the TEAD4–VGLL1 and TEAD4–YAP complexes use their distinctive transcriptional machineries.

Several inhibitors of the YAP–TEAD4 complex binding have been reported. Flufenamic acid, which binds the central pocket of the YAP-binding domain of TEAD2, inhibits the proliferation and migration of cancer cell [22]. Verteporfin, which disrupts the YAP–TEAD complex, is found to increase the sensitivity to paclitaxel in HCT-8/T cells [30], as well as sensitivity to erlotinib in lung cancer cells [31]. Likewise, the disruption of TEAD-VGLL1 interactions might be of use in development of anticancer drugs.

In this study, we discovered *VGLL1* as a novel prognostic biomarker correlated with *PIK3CA* or *PIK3CB* in gastric cancer. We elucidated the molecular mechanism underlying the regulation of *VGLL1* transcription by the PI3K/AKT/β-catenin. The formation of the VGLL1–TEAD4 complex activates the transcription of *MMP9*, which then promotes proliferation and metastasis in gastric cancer cells. Taken together, we clearly elucidated the molecular mechanism that involves *VGLL1* that promotes malignancy in gastric cancer, thus suggesting *VGLL1* as a therapeutic target in treatment of gastric cancer.

## 4. Materials and Methods

### 4.1. Reagents

U0126, LY294002, KN93, SP600125, were purchased from Sigma-Aldrich (St. Louis, MO, USA). TGF-β was purchased from Peprotech (Seoul, Korea).

### 4.2. Cell Culture

Human gastric cancer cells (NUGC3, AGS, NCI-N87, SNU1, SNU5, SNU16, SNU216, SNU484, SNU668, Hs746T, and MKN28 cells) were cultured in RPMI-1640 medium that contained 10% fetal bovine serum (FBS). NUGC3 cells constitutively expressing *VGLL1* (NUGC3-*VGLL1* cells) were selected with 500 μg/mL geneticin (Thermo Scientific, Logan, UT, USA). HEK293T cells were cultured in Dulbecco’s modified Eagle’s medium (DMEM) containing 10% FBS. All the cells were cultured in an atmosphere of 5% CO_2_ at 37 °C. STR profiling by Korea Cell Line Bank authenticated all of the cell lines (Seoul, Korea).

### 4.3. Patients

A retrospective review of a gastric cancer cohort database prospectively maintained at Yonsei University College of Medicine (Seoul, South Korea) was conducted to identify all the gastric adenocarcinoma patients who underwent curative D2 gastrectomy between 2000 and 2010. Demographic and clinicopathological information and tumor tissue samples were obtained from 556 patients. The institutional review board of Severance Hospital approved this study (Seoul, Korea; 2015-3104-001).

### 4.4. Microarray Experiments and Data Processing

The total RNA extracted from 556 gastric cancer tissues was used for labeling and hybridization, according to the manufacturer’s protocols (Illumina HumanWG-6 BeadChip, version 2, Illumina, San Diego, CA, USA). The arrays were scanned with an Illumina Bead Array Reader confocal scanner (BeadStation 500GXDW; Illumina, Inc., San Diego, CA, USA), as per the manufacturer’s instructions. After scanning, the microarray data were log2 transformed, median centered across genes and samples, and normalized while using quantile normalization in the R language environment (version 3.2.5, The R Foundation for Statistical Computing, Vienna, Austria). The microarray data set of gastric cancer samples from patients is available in the NCBI Database of GEO datasets under the data series accession numbers GSE13861 and GSE84437. The microarray analysis of *VGLL1* siRNA-treated NUGC3 cells while using same procedure and microarray data set is available in the NCBI Database of GEO datasets under accession numbers GSE130071. We applied the FDR approach for analysis of microarray data. After FDR correction, we selected genes (*p* value < 0.05 and log_2_FC > 1) for GO analysis.

### 4.5. Statistical Analysis of Microarray

Pearson correlation coefficients were calculated for evaluating the association between genes. We obtained an optimal cut-off for gene expression from ROC analysis to classify patients into two subgroups by single gene expression, in which the best cut-off was determined by the expression with the highest multiply of sensitivity and specificity. Statistical analyses was carried out while using Medcalc version 18.11.6 (MedCalc software, Ostend, Belgium). The Kaplan-Meier method was used to calculate the time before death or recurrence, and difference between the times was assessed using logrank test (MedCalc software, Ostend, Belgium). A gene set enrichment analysis was carried out to identify the most significant gene sets associated with molecular and cellular functions. Fisher’s exact test estimated the significance of over-represented gene sets. Gene set enrichment analyses were performed using the DAVID bioinformatics resources (ver. 6.8, Laboratory of Human Retrovirology and Immunoinformatics, Frederick, MD, USA).

### 4.6. Plasmids Construction

pOBT7-*VGLL1* and pCMV-SPORT6-*TEAD4* were obtained from Korean UniGene Information (KUGI). *VGLL1* mRNA was amplified via PCR and then cloned into the HindIII/BamHI sites of pcDNA3.1 with Myc-tag. *TEAD4* was inserted in the HindIII/BamHI sites of pEGFP-N1. *YAP* was amplified via PCR and then cloned into the EcoRV/XbaI sites of pcDNA3.1 with Myc-tag. Fragments of the *MMP9* promoter (−657/+25, −557/+25) were PCR-amplified and subsequently inserted into the KpnI/XhoI site of pGL4.17 (Luc2/neo). The *MMP9* promoter, which contained mutation in TEA binding site (5′-CATTCC-3′→5′-CAGGGC-3′), was generated while using the Quikchange Site-Directed Mutagenesis kit. The fragment of VGLL1 promoter was PCR-amplified and cloned into the XhoI/HindIII site of pGL2-basic luciferase plasmid. pCMV3-β-catenin, pCMV3-*p300*, pCMV3-*TCF4*, and pCMV3-*LEF1* were obtained from Sino Biological (Wayne, PA, USA).

### 4.7. Live cell Assay for Cell Proliferation and Migration

Proliferation rates, which were based on cell confluence, were determined by live-cell imaging (IncuCyte ZOOM System, Essen BioScience, Ann Arbor, MI, USA), as described previously [32]. To analyze cell migration, the cells were cultured in 96-well ImageLock Plates (Essen BioScience) to reach confluence prior to wound creation. A scratch was made in confluent monolayers while using a 96-pin WoundMaker (Essen BioScience, Ann Arbor, MI, USA). The cells were washed with PBS and then incubated using the IncuCyte ZOOM system. Cell migration was analyzed at 2 h intervals throughout the duration of the experiment.

### 4.8. Spheroid Formation

Three-dimensional (3D)-Spheroid culture was induced, as described previously [33]. In brief, cells were trypsinized, counted, and diluted to 5 × 10^4^ cells in a 20 μL droplet. The droplets of cell suspension were placed on the lid of a sterile non-adherent polystyrene petri dish that was filled with DPBS, and then cultured at 37 °C in a 5 % CO_2_ incubator for 48 h.

### 4.9. Mice Experiments

The bioethics committee of the Korea Research Institute of Bioscience and Biotechnology approved all animal experiments (KRIBB-ACE-16101, KRIBB-ACE-17051, and KRIBB-ACE-18209). In vivo xenografts were performed, as described previously [32]. NUGC3 cells (5 × 10^6^) that were infected with a lentiviral sh*VGLL1* vector were subcutaneously injected into five-week-old female BALB/c nude mice. According to the protocol that was published by Le A. [34], stable *VGLL1*-expressing HEK293T cells (1 × 10^7^) were subcutaneously injected into five-week-old female BALB/c nude mice.

We established an in vivo xenograft mouse model to study the effect of VGLL1 on gastric cancer metastasis. Five-week-old female nude mice (six mice per group) were subcutaneously inoculated with NUGC3-EV or NUGC3-*VGLL1* cells (5 × 10^6^) in the right flank. Surgical resection of the primary tumor was performe when the average tumor volume reached 500 mm^3^. After four weeks, the mice were sacrificed by cervical dislocation under isoflurane anesthesia, and lung and liver tissue was also removed for subsequent experiments. For a tail vein-injection assay of cancer metastasis, NUGC3 cells (1 × 10^6^) cells that were infected with a lentivirus expressing shControl or sh*VGLL1* were injected into the tail vein of mice (four mice per group) in 100 µL of phosphate-buffered saline (PBS). After 16 weeks, the lungs were removed and fixed. Hematoxylin and eosin (H&E) assessed tumor metastasis to the lungs and liver. Photos of random fields were obtained at a magnification of 40× (3 fields/mouse) and analyzed while using NIH Image software for the quantitation of metastasis (ver. 1.48, Wayne Rasband, Bethesda, Maryland, USA) [35].

### 4.10. Western Blot Analysis

The cells were lysed with RIPA buffer (Millipore, Billerica, MA, USA) containing protease inhibitor cocktail (Roche), and the lysates were quantified with a protein assay kit (Bio-Rad, Hercules, CA, USA). The cell lysates were separated using SDS-PAGE and then transferred to PVDF membranes. The proteins were identified while using appropriate antibodies. Anti-VGLL1 (10124-2-AP) was purchased from Proteintech (Rosemont, IL, USA). Anti-TEAD4 (ab58310) and anti-MMP9 (ab76003) were purchased from Abcam (Cambrige, MA, USA). Anti-GFP (NB600-308) was purchased from Novus Biologicals (Centennial, CO, USA). Anti-Myc (sc-789) and anti-HA (sc-805) were purchased from Santa Cruz Biotechnology (Dallas, TX, USA). Anti-GAPDH (LF-PA0212) was purchased from AbFrontier (Seoul, Korea). Anti-β-tubulin (2128), anti-pSer473-AKT (9271), anti-AKT (9272), anti-p-β-catenin (9561), and anti-β-catenin (9562) were purchased from Cell Signaling (Danvers, MA, USA). Anti-Flag (F1804) was purchased from Sigma-Aldrich (St. Louis, MO, USA). The protein signal was detected while using an enhanced chemiluminescence kit (Millipore, Burlington, MA, USA).

### 4.11. Reverse Transcriptase Polymerase Chain Reaction and Quantitative Real-Time PCR

The total RNA was isolated using TRIzol reagent (Invitrogen, Carlsbad, CA, USA), and cDNA was synthesized using TOPscript™ RT DryMIX (Enzynomics, Daejeon, Korea). RT-PCR was performed using Dr. Taq MasterMix (Doctor Protein, Daejeon, Korea). Quantitative real-time PCR was performed using a SYBR Green master mix kit (Qiagen, Valencia, CA, USA). The following sequences of primers were used: *VGLL1* (F) 5′-GAGCTGTGGCATTTCTCCTC-3′, (R) 5′-AAGTGGGTGTGAGCAGCTTT-3′, *MMP9* (F) 5′-TCTATGGTCCTCGCCCTGAA-3′, (R) 5′-CATCGTCCACCGGACTCAAA-3′, *TEAD4* (F) 5′-GAACGGGGACCCTCCAATG-3′, (R) 5′-GCGAGCATACTCTGTCTCAAC-3′, *YAP* (F) 5′-CGCTCTTCAACGCCGTCA-3′, (R) 5′-AGTACTGGCCTGTCGGGAGT-3′, *CTGF* (F) 5′-CTTGCGAAGCTGACCTGGAA-3′, (R) 5′-AAAGCTCAAACTTGATAGGCTTGGA-3′, *PIK3CA* (F) 5′-TGCAGCCATTGACCTGTTTA-3′, (R) 5′-GTCAAAACAAATGGCACACG-3′, *GAPDH* (F) 5′-TCATGACCACAGTCCATGCC-3′, (R) 5′-TCCACCACCCTGTTGCTGTA-3′, *RPL13A* (F) 5′-CATCGTGGCTAAACAGGTAC-3′, and (R) 5′-GCACGACCTTGAGGGCAGC-3′. The primers of *PIK3CB* (P144505) was purchased from Bioneer (Daejeon, Korea).

### 4.12. Gene Knockdown Using siRNA

Introducing siRNA into the target gene, using Lipofectamine 2000 (Invitrogen, Carlsbad, CA, USA), according to the manufacturer’s instructions, was undertaken to perform gene knockdown. The siRNA sequences were, as follows: siScramble (siSC) 5′-CCUACGCCACCAAUUUCGUdTdT-3′, si*VGLL1* 5′-AGCCUAUAAAGACGGAAUGGA AUdTdT-3′ and 5′-CCCGGUGUGUCCUUUUCACCUACdTdT-3′, si*TEAD4* 5′-GACACUACUCUUACCGCAUdTdT-3′, si*YAP* 5′-CCACCA AGCUAGAUAAAGAAAdTdT-3′, si*MMP9* 5′-CCACAACAUCACCUAUUGGAUdTdT-3′, si*p300* 5′-GAUGAAUGCGGGCAUGAAU dTdT-3′, siβ-*catenin* 5′-CGUUCUCCUCAGAUGGUGUdTdT-3′, siTCF4 5′-CAGACAAAGAAAGUUCGAAdTdT-3′, si*LEF1* 5′-GAACGACUCUGAAAUCAUCUU dTdT-3′, si*PIK3CA* 5′-CUGAGAAAAUGAAAGCUCACUCUdTdT-3′, and si*PIK3CB* 5′-CAGUACAAUUUGGUGUCAUdTdT-3′.

### 4.13. Lentivirus Infection

shControl and sh*VGLL1*(Sigma, TRCN0000019618) were packaged into lentivirus via HEK293T cells, using Lipofectamine and PLUS Reagent (Invitrogen, Carlsbad, CA, USA), and then transduced into NUGC3 cells. Post-transduction 48 h, the NUGC3 cells were selected with puromycin (1 μg/mL).

### 4.14. Immunohistochemistry (IHC)

US Biomax supplied tissue array blocks of human gastric cancer and normal tissues (Rockville, MD, USA). IHC was performed, as previously described [32]. The slides were incubated with anti-VGLL1 (Proteintech, 10124-2-AP, Rosemont, IL, USA), anti-Ki67 (abcam, ab15580), and anti-MMP9 (abcam, ab76003) antibodies. After washing with PBS, the slides were incubated with biotinylated anti-rabbit IgG (Vector Laboratories, Burlingame, CA, USA) and avidin-biotin peroxidase (Vector Laboratories) and visualized using diaminobenzidine tetrahydrochloride (Vector Laboratories, Burlingame, CA, USA). The sections were counterstained with hematoxylin.

### 4.15. Invasion Assay

For invasion assays, chambers with 8.0-μm-pore PET membrane in 24-well cell culture inserts (BD Biosciences, San Jose, CA, USA) were used. The cells in serum-free medium were seeded into the upper part of each chamber with Matrigel coating, whereas the lower compartments were filled with the above-mentioned medium. The cells were then allowed to invade, being subsequently fixed with 10 % formalin, and stained with sulforhodamine B (SRB), as previously described [36].

### 4.16. Luciferase Assay

A dual-luciferase reporter system determined the promoter activity (Promega, Madison, WI, USA). The cells were transfected with pGL4.17-*MMP9*-luciferase (*MMP9*-luc), pGL2-*VGLL1*-luciferase (*VGLL1*-luc), and pRL-SV40 plasmid encoding firefly (Renilla)-luciferase, using PolyFect (Qiagen, Valencia, CA, USA). The luciferase activity was measured using a luminometer (VICTOR X Light; PerkinElmer, Waltham, MA, USA). The results were normalized to the activity of Renilla luciferase.

### 4.17. Chromatin Immunoprecipitation (ChIP) Assays

ChIP assays were performed, as previously described [36]. Briefly, the cells were crosslinked using 1% formaldehyde and then lysed. The chromatin was then sheared via sonication on ice. The lysates were incubated with antibodies targeting β-catenin, VGLL1 or TEAD4, or with normal mouse immunoglobulin G (IgG) overnight at 4 °C. Thereafter, protein was digested using proteinase K (Millipore, Middlesex County, MA, USA). The ChIP-enriched DNA was subjected to PCR while using either of the following two primers: VGLL1-a (5′-GTA GAC AAA GAG AGG AGC-3′ and 5′-GGC TTC CAT TGG CCA AAG-3′), VGLL1-b (5′-TTT GTT GTT GAC TCT GTG T-3′ and 5′-AAG GCG TTT CCT GCT AGC-3′), MMP9-a (5′-TACTGTCCCCTTTACTGC-3′ and 5′-CTTCCTCTCCCTGCTTCA-3′), and MMP9-b (5′-TGGTGTAAGCCCTTTCTC-3′ and 5′-AGGAGGCGCTCCTGTGAC-3′).

### 4.18. Statistical Analyses

Student’s *t*-tests or Chi-square tests were used for statistical analyses. The bars indicate S.D., and the asterisks denote significant differences (*** *p* ≤ 0.005, ** *p* ≤ 0.01, * *p* ≤ 0.05) between the means of two groups.

## 5. Conclusions

*VGLL1* is a prognostic biomarker that correlates with *PIK3CA* and *PIK3CB* in gastric cancer. PI3K/AKT/β-catenin transcriptionally activates *VGLL1*, thus calibrating *MMP9* expression linked to gastric cancer malignancy. Our finding proposes *VGLL1* as a therapeutic target for malignant gastric cancer.

## Figures and Tables

**Figure 1 cancers-11-01923-f001:**
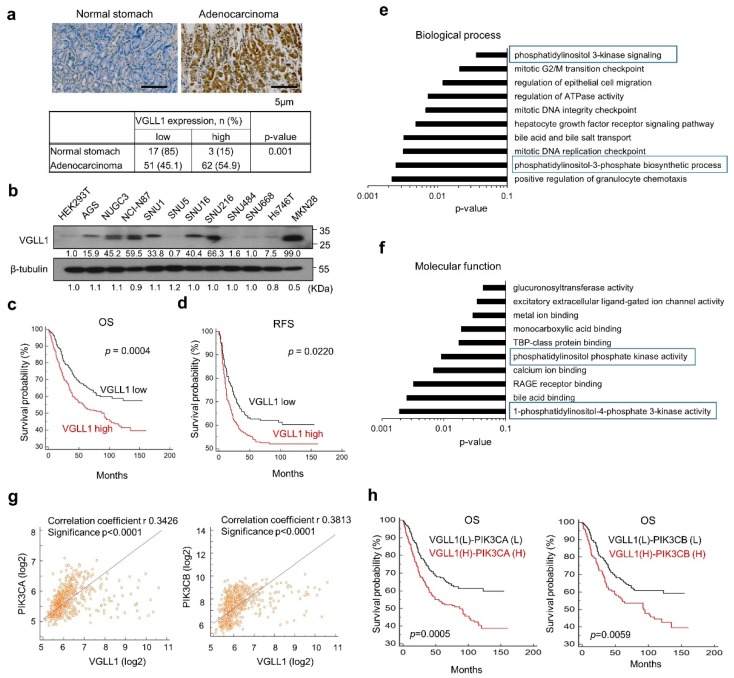
Vestigial-like 1 (VGLL1) expression is correlated with gastric cancer and PI3K. (**a**) VGLL1 expression in gastric cancer. Representative IHC images of normal and gastric cancer tissue samples (original magnification: 200×). Scale bar: 5 µm. Chi-squared test. (**b**) VGLL1 expression in human gastric cancer cell lines was analyzed using western blotting. (**c**,**d**) Kaplan-Meier curves of overall survival (OS) and recurrence-free survival (RFS) in gastric cancer patients stratified by VGLL1 expression. Survival curves were compared using Log rank test. (**e**,**f**) In Microarray analysis of 556 patients with gastric cancer, 202 genes were up-regulated in VGLL1 high subgroup compared to VGLL1 low subgroups (FDR correction, *p* value < 0.05 and log_2_FC > 1). For Gene Ontology analysis of the 202 genes, classification enrichment was determined while using the DAVID tool. (**g**) Pearson correlations between VGLL1 and PIK3CA or PPIK3CB in gastric cancer patients. (**h**) Kaplan-Meier curves of gastric cancer patient subgroups defined by their combination of VGLL1 and PIK3CA or PPIK3CB expression levels.

**Figure 2 cancers-11-01923-f002:**
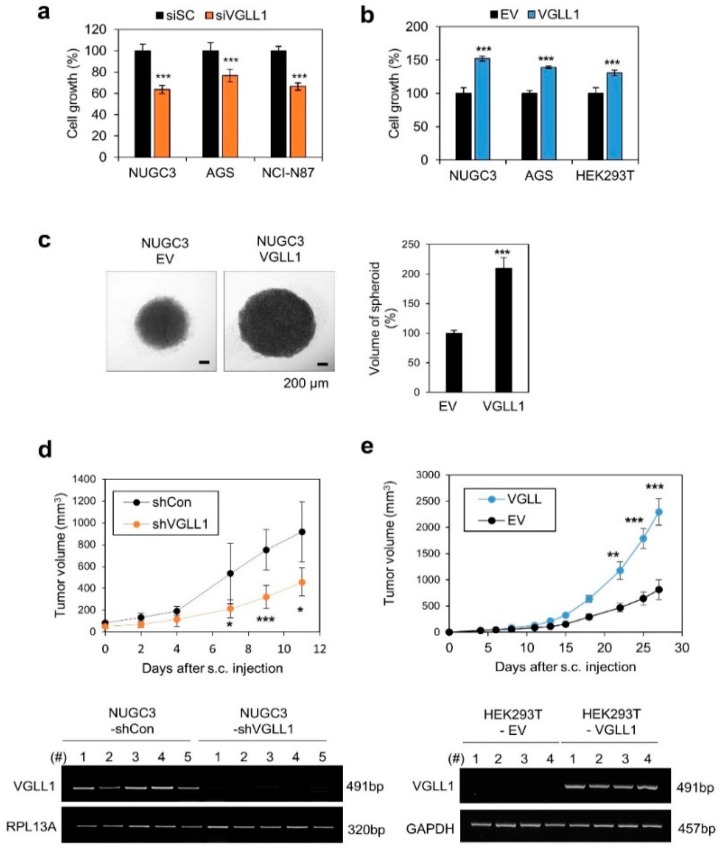
VGLL1 is involved in gastric cancer cell proliferation. (**a**,**b**) Effect of VGLL1 expression knockdown (a) or overexpression (b) on cell growth measured by live-cell imaging. *n* = 4; *** *p* < 0.005 (Student’s *t*-test). (**c**) Effect of *VGLL1* overexpression on spheroid formation in stable *VGLL1*-overexpressing NUGC3 (NUGC3-VGLL1) and control (NUGC3-Empty Vector) cells. *n* = 5; *** *p* < 0.005 (Student’s *t*-test). (**d**) In vivo tumor formation of NUGC3 cells infected with Lenti-sh*VGLL1* or Lenti-shControl. (**e**) In vivo tumor formation of *VGLL1*-overexpressing HEK293T cells. VGLL1 expression in tumor tissues of the xenograft model as measured by RT-PCR. * *p* < 0.05, ** *p* < 0.01, *** *p* < 0.005 (Student’s *t*-test).

**Figure 3 cancers-11-01923-f003:**
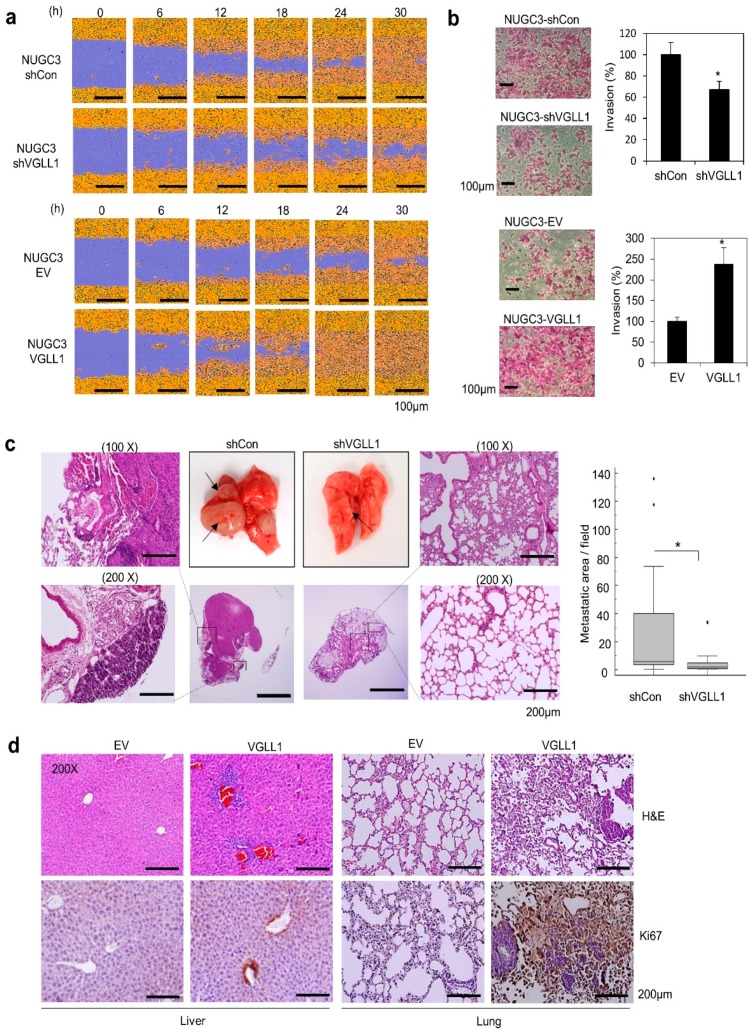
*VGLL1* is involved in gastric cancer metastasis. (**a**) Wound healing assay with *VGLL1* overexpression or knockdown in NUGC3 cells. Representative images of scratch wounds at the indicated times. (**b**) Invasion assays with *VGLL1*-expressing or knocked down NUGC3 cells incubated for 40 or 48 h, respectively. *n* = 3; * *p* < 0.05 (Student’s *t*-test). (**c**) Tail vein injection assay of NUGC3 cells expressing sh*VGLL1*. H&E staining images of the lungs removed at 16 weeks after injection. The graph indicates the size of metastatic lesions (*n* = 12). Scale bar, 200 µm. * *p* < 0.05 (Student’s *t*-test). (**d**) In vivo metastasis assay by surgical resection of *VGLL1*-expressing NUGC3 cell tumors. The liver and lungs obtained from mice sacrificed at 29 days after surgical resection of tumors formed using VGLL1-expressing NUGC3 cells. H&E staining and IHC staining for Ki-67 in the liver and lungs. Scale bar, 200 µm.

**Figure 4 cancers-11-01923-f004:**
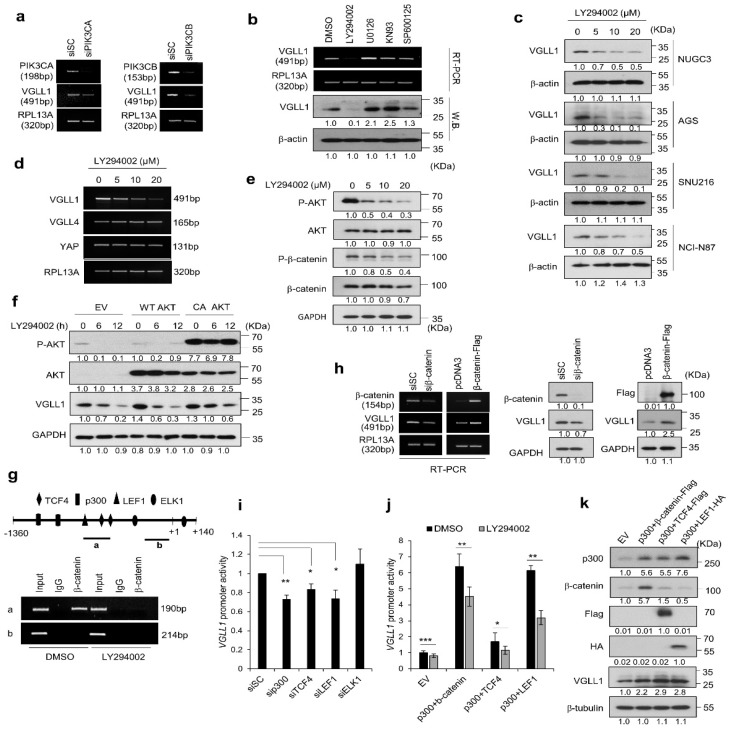
*VGLL1* expression is regulated by the PI3K/Akt/β-catenin pathway. (**a**) Expression of *VGLL1* mRNA after treatment with si*PIK3CA* or si*PIK3CB*. (**b**) NUGC3 cells incubated with inhibitors (10 µM) for 24 h. The mRNA and protein expression of VGLL1 was analyzed by RT-PCR and western blotting, respectively. (**c**) Effects of LY294002 on VGLL1 expression in gastric cancer cells. (**d**) mRNA levels of *VGLL1*, *VGLL4*, and *YAP* in NUGC3 cells treated with LY294002 and analyzed by RT-PCR. (**e**) NUGC3 cells incubated with LY294002 for 6 h at the indicated concentrations. Expression of the proteins was analyzed by western blotting. (**f**) Effect of AKT on VGLL1 expression assessed by western blotting. NUGC3 cells expressing wild-type (WT) AKT or constitutively active (CA) AKT were treated with LY294002. (**g**) Localization of β-catenin to the VGLL1 promoter. ChIP assay was performed using nuclear extracts of NUGC3 cells treated with LY294002. The ChIP-enriched DNA was subjected to PCR. (**h**) Effect of β-catenin expression knockdown or overexpression on *VGLL1* expression in NUGC3 cells analyzed by RT-PCR and western blotting. (**i**) *VGLL1*-promoter-driven luciferase reporter activity. NUGC3 cells were treated with siRNAs and then transfected with VGLL1-luc and Renilla-luc vectors. (**j**) Effect of LY294002 on VGLL1 promoter activity. NUGC3 cells were co-transfected with β-catenin, *p300*, *TCF4*, *LEF1*, *VGLL1*-luc, and Renilla-luc for 24 h and then incubated with 10 µM LY294002 for 24 h. Mean ±SD of three independent experiments with triplicate measurements.* *p* < 0.05, ** *p* < 0.01, *** *p* < 0.005 (Student’s *t*-test). (**k**) The effects of β-catenin, p300, TCF4, and LEF1 on VGLL1 expression, assessed by western blotting.

**Figure 5 cancers-11-01923-f005:**
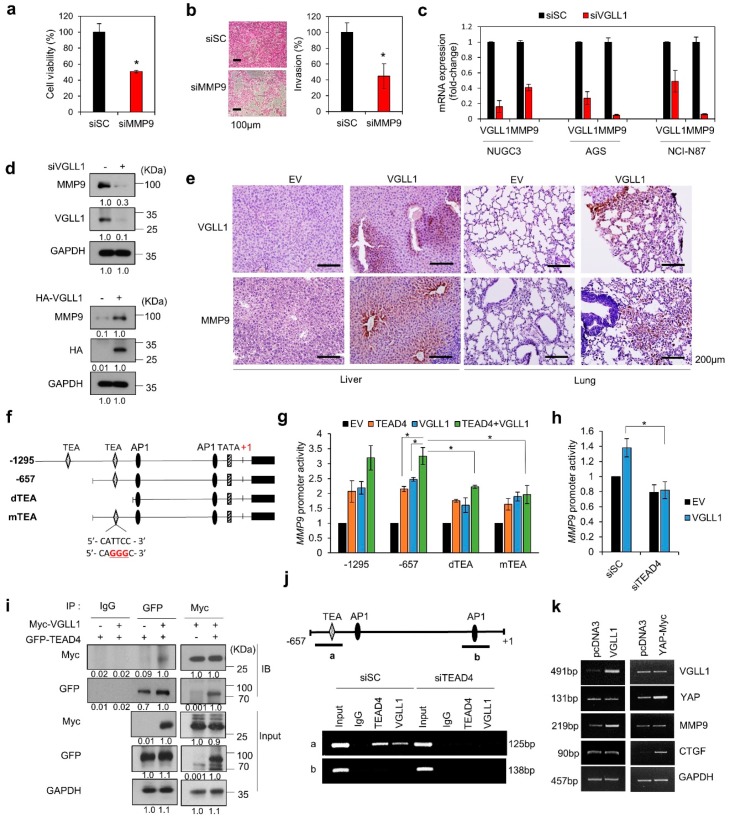
VGLL1 regulates *MMP9* transcription in gastric cancer cells. (**a**,**b**) Viability (**a**) and invasion (**b**) assay of NUGC3 cells. Cells were treated with 20 nM siScramble (SC) or si*MMP9* for 48 h, and then stained with sulforhodamine B. Data are presented as mean ±SD. *n* = 3; * *p* < 0.05 (Student’s *t*-test). (**c**) *MMP9* mRNA expression regulation by VGLL1 in gastric cancer cells treated with si*VGLL1* or siSC (control) was measured by qPCR. (**d**) Changes in MMP9 expression upon knockdown or overexpression of *VGLL1* assayed by western blotting in NUGC3 cells. (**e**) IHC of VGLL1 and MMP9 expression in the liver and lungs of an in vivo metastasis mouse model using surgical resection of tumors. Scale bar, 200 µm. (**f**) Construction of various luciferase reporter systems under control of the *MMP9* promoter. (**g**) *MMP9* promoter activities of the reporter systems containing modified TEA-binding sites were measured in NUGC3 cells. *n* = 3; * *p* < 0.05 (Student’s *t*-test). (**h**) Effect of TEAD4 on VGLL1-regulated *MMP9* transcriptional activity. NUGC3 cells treated with siSC or si*TEAD4* for 24 h were transfected with *MMP9*-luc, Renilla-luc, pCDNA3.1, and pcDNA3.1-myc-*VGLL1* vectors for 48 h. *n* = 3; * *p* < 0.05 (Student’s *t*-test). (**i**) Interaction between VGLL1 and TEAD4. Lysates of NUGC3 cells that were transfected with pcDNA3.1-myc-*VGLL1* and pEGFP-N1-*TEAD4* were immunoprecipitated using anti-IgG, anti-GFP, and anti-Myc antibodies. Protein expression was analyzed by immunoblotting. (**j**) ChIP assays while using nuclear extracts of NUGC3 cells treated with si*TEAD4*. The ChIP-enriched DNA was subjected to PCR. (**k**) Target genes of VGLL1 and YAP. NUGC3 cells were transfected with pcDNA3, pcDNA3-myc-*VGLL1*, or pcDNA3-myc-*YAP*, and the mRNA expression levels were analyzed by RT-PCR.

**Figure 6 cancers-11-01923-f006:**
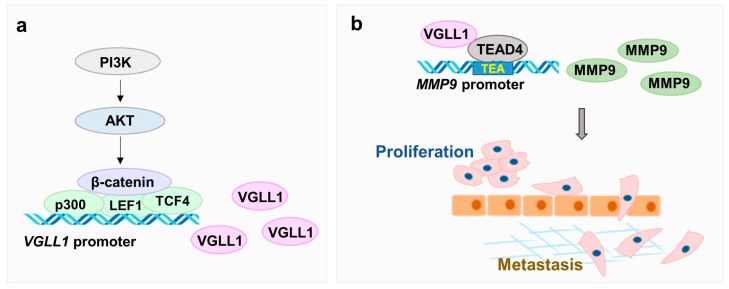
Cell proliferation and metastasis of gastric cancer by modulation of VGLL1 expression and activity. (**a**) Transcription of VGLL1 is regulated by the PI3K/AKT/β-catenin pathway. (**b**) VGLL1 formed complex with TEAD4 to activate transcription of MMP9, a target gene of VGLL1.

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
