# Peer review of "PI3K/AKT/β-Catenin Signaling Regulates Vestigial-Like 1 Which Predicts Poor Prognosis and Enhances Malignant Phenotype in Gastric Cancer"

_cancers, 2019, doi:10.3390/cancers11121923_

Round 1

Reviewer 1 Report

PI3K/AKT/β-catenin signaling regulates Vestigial-2 like 1 which predicts poor prognosis and enhances 3 malignant phenotype in gastric cancer

Kim et al present an interesting and potentially therapeutically important discovery that Vestigial-like 1 (VGLL1) is involved in gastric cancer proliferation and metastasis. The authors posit that VGLL1 is regulated by PI3K/AKT/beta-catenin activity and that VGLL1 regulates metastasis through MMP9 expression. The paper is well written and generally includes appropriate controls to support the authors’ conclusions. There are, however, several points which should be clarified or supported with further experiments, as detailed below. If the authors can satisfy these concerns, the paper would be appropriate for publication in Cancers.

Major points:

- Page 2, Line 54: the authors say that TCGA has identified four sub-types of gastric cancer, but they only describe three in the paragraph: CIN, MSI, EBV. What is the fourth type?

- The title for Figure 1 is “Figure 1. VGLL1, a prognostic biomarker, is crucial for the proliferation of gastric cancer cells.” This title is not supported by the data. The data in this figure shows that VGLL1 expression is correlated with gastric cancer and PI3K.

- Figure 1G: are these Pearson correlations or Spearman correlations? It appears from the methods that these are Pearson correlations, but the authors should explicitly mention this in the legend.

- In Figure 1, is VGLL1 more upregulated in any of the four types of gastric cancer? For example, since the EBV sub-type is more likely to have PI3K mutations, is more VGLL1 expression observed in these cancers? The authors should include this information about VGLL1 by sub-type (assuming the sub-type for their tumors measured by microarray is known).

- Page 11, “Microarray analysis” Have the authors applied an FDR correction to their significance test? P < 0.001 is likely not significant for microarray with >10k genes.

- Overexpression of VGLL1 in NUGC3 cells. From Figure 1B, NUGC3 cells already overexpress VGLL1. Why have these cells been chosen as the model cell line for overexpression of VGLL1 in Figures 2C and 3? NUGC3 seems an appropriate model cell line for the shRNA knockdown experiments, but the authors should show the effects of VGLL1 overexpression in non-VGLL1 overexpressing gastric cell lines.

- Figure 2E: why are the authors using HEK293T cells with VGLL1 overexpression (and not a gastric cancer cell line)? They have VGLL1 overexpressing gastric cancer lines (notably NUGC3 cells, which are used in Fig. 2C, although as described in my previous comment these are not the ideal model for VGLL1 overexpression). The authors need to either justify the switch to HEK293T or re-do the experiment with a gastric cancer cell line.

- Figure 4E-F: The authors show “P-AKT” Western blotting. What phosphorylation site is being tested here? The authors list the product number Cell Signaling 92271. I looked on the CST website and couldn’t find this antibody. I think the authors are referring to product 9271 (not 92271) which is pSer473-AKT, but they should clarify.

- Figure 4F: The authors use “constitutively active AKT” but don’t describe what is this protein. Point mutation? Myristoylation? I also don’t see this vector described in the methods. The authors need to clarify.

- Figure 4H: The authors have overexpressed b-catenin-FLAG in a cell line (though it’s not clearly stated which cell line in the legend) that appears to express endogenous b-catenin (as shown by the siControl lane at left). Why don’t the RT-PCR and Western blot at right with b-catenin FLAG show endogenous expression of b-catenin in the pcDNA3 lanes? Is the overexpression so substantial that the endogenous mRNA / protein are not detectable at these exposures?

- Figure 4: the effects of b-catenin knockdown and overexpression on VGLL1 levels are rather modest (Fig. 4I, for example) and the data for b-catenin’s role is all indirect (knockdown of TCF4 or LEF1, for example). Can the authors show a ChIP demonstrating that b-catenin is localized to the VGLL1 promoter? Can the authors show that PI3K knockdown or LY294002 treatment reduces the amount of b-catenin at the VGLL1 promoter? This data is needed to convincingly show that b-catenin is what is driving VGLL1 expression in gastric cancer. It could well be that PI3K/AKT are working through another mechanism to regulate VGLL1 transcription.

Minor points:

- Page 2, Line 59, This sentence is repeated: “The Epstein-Barr virus (EBV) subtype has a high rate of PIK3CA mutations and 59 high PD-L1/L2 expression.”

- Page 4, Line 116: “contro” should be “control”

Reviewer 2 Report

Minor comments:

Line 35: the word "by" is redudant

Line 116: letter "l" is missing from the word "control"

Figure 3a: Time points for left and right panels completely arbitrary 

Reviewer 3 Report

The authors demonstrated the oncogenic role of VGLL1 in gastric carcinogenesis. The found VGLL1 is positively regulated by PI3K/Akt/beta-catenin signaling. Through binding with TEAD transcription factor, VGLL1 activates the transcription of MMP9, thus to promote GC cell metastasis. The concluded VGLL1 serves as a prognostic biomarker and potential therapeutic target. In general, their findings are novel and the results are solidly generated. Major concerns are as follows:

The authors employed their own cohort for correlation analysis such as in Figure 1g. It is better to use other large-scale cohort for validation such as TCGA to get a solid conclusion. As the same, is any positive correlation observed between VGLL1 and MMP9 in primary samples? In Figure 2 and 3, the authors simply performed some functional tests, however several issues needed to be added. For example, it is better to confirm the VGLL1 expression level by Western blot or qRT-PCR in the xenografts to make sure the in vivo data are reliable. As VGLL1 regulates MMP9 expression through TEAD4. It is better to demonstrate if knocking out TEAD4 abolishes the regulatory effect. Any co-IP data supporting the direct interaction between VGLL1 and TEAD4?

Round 2

Reviewer 1 Report

PI3K/AKT/β-catenin signaling regulates Vestigial-2 like 1 which predicts poor prognosis and enhances 3 malignant phenotype in gastric cancer

Manuscript ID: cancers-638196

The authors have done a reasonable job responding to my initial comments. However, there are still some points that need to be better addressed before this is suitable for publication in Cancers.

Major points:

My original comment:

“Microarray analysis” Have the authors applied an FDR correction to their significance test? P < 0.001 is likely not significant for microarray with >10k genes.

Authors response:

In microarray analysis, FDR correction can increase statistical significance, but we did not applied to select only those with biological relevance. The analysis was performed according to the data processing described in Methods 4.4. Methods 4.5 (page 11) analyzed p <0.001 as significant in statistical analysis such as comparison test and Pearson correlation based on microarray results. We deleted the part because it seems to confuse the contents.

My revised comment:

The authors are simply wrong to ignore FDR correction. The legend of Figure 1 says “176 genes were up-regulated in VGLL1 high subgroup compared to VGLL1 low subgroups (P<0.001, fold change>1)”. I have no confidence that those genes are actually “upregulated” if the authors cannot or are not willing to do an FDR correction. If the GO analysis after FDR correction no longer supports their conclusions, it would be better to remove it completely from the manuscript than to present incorrect analysis.

My original comment:

Overexpression of VGLL1 in NUGC3 cells. From Figure 1B, NUGC3 cells already overexpress VGLL1. Why have these cells been chosen as the model cell line for overexpression of VGLL1 in Figures 2C and 3? NUGC3 seems an appropriate model cell line for the shRNA knockdown experiments, but the authors should show the effects of VGLL1 overexpression in non-VGLL1 overexpressing gastric cell lines.

Authors response:

We appreciate your valuable comments. In Figure 2b, we showed increased proliferation of ACS and HEK293T cells which expressing relatively low VGLL1 (refer to Figure 1b). We further examined the effect of VGLL1 overexpression on cell proliferation in SNU484, SNU638 and SNU668 cells, which express VGLL1 at low level. Cells were transfected with pcDNA3.1 and pcDNA3.1–VGLL1 for 72 h, and then stained with sulforhodamine B. Obviously, VGLL1 overexpression increased the proliferation of gastric cancer cells tested.

My revised comment:

Thank you for this additional data. From the manuscript text, it appears that this data was supposed to be inserted as Supplementary Material Figure S1b,c. However, in the version of Supplementary Material that was included, Figure S1b,c is Western blotting, not growth data. I recommend that you add the growth data with these three new cell lines to the manuscript to further support your findings.

My original comment:

Figure 2E: why are the authors using HEK293T cells with VGLL1 overexpression (and not a gastric cancer cell line)? They have VGLL1 overexpressing gastric cancer lines (notably NUGC3 cells, which are used in Fig. 2C, although as described in my previous comment these are not the ideal model for VGLL1 overexpression). The authors need to either justify the switch to HEK293T or re-do the experiment with a gastric cancer cell line.

Authors response:

We appreciate your valuable comment. We wanted to observe the effect of VGLL1 overexpression in normal cells expressing low VGLL1. However, we could not obtain normal gastric cancer cells. Therefore, we selected HEK293T cells because it is close to normal cells and expresses low level of VGLL1. We confirmed the increase of proliferation by VGLL1 overexpression in HEK293T cells (Figure 2b).

My revised comment:

My comment was not about the effect of VGLL1 in normal gastric cell lines. It was rather about the effect of VGLL1 overexpression on gastric cancer cell line xenografts. The authors have extensive data about the effects of VGLL1 overexpression on the proliferation of gastric cancer cell lines in 2D. However, for the xenografts, they have only HEK293T cells (no gastric cancer cell line xenografts). If the authors cannot or are not willing to re-do the xenograft experiment with an actual gastric cancer cell line, then they should at least qualify their data with clear statements that although HEK293T are not gastric cancer cells, these results are suggestive but do not prove that VGLL1 is important for increased proliferation of xenografts.

Reviewer 3 Report

I think the authors have addressed all my concerns. No other issues raised.

Author Response

We appreciate your interest and valuable comments on our manuscript.

Thanks again.

Sincerely,

Round 3

Reviewer 1 Report

The authors have sufficiently addressed all my concerns, and I recommend the manuscript for acceptance.